# Do the Fastest Open-Water Swimmers have A Higher Speed in Middle- and Long-Distance Pool Swimming Events?

**DOI:** 10.3390/jfmk4010015

**Published:** 2019-03-20

**Authors:** Roberto Baldassarre, Maddalena Pennacchi, Antonio La Torre, Marco Bonifazi, Maria Francesca Piacentini

**Affiliations:** 1Department of Movement, Human and Health Sciences, University of Rome Foro Italico, 00135 Rome, Italy; 2Department of Biomedical Sciences for Health, University of Milan, 20133 Milan, Italy; 3IRCCS Istituto Ortopedico Galeazzi, 20161 Milan, Italy; 4Department of Medicine, Surgery and Neuroscience, University of Siena, 53100 Siena, Italy

**Keywords:** endurance performance, elite athletes, discriminant analysis

## Abstract

Background: It has been shown that the fastest open-water swimmers (OW-swimmers) increase significantly the speed in the last split of the open-water events. The aim of the present work was to determine if the fastest OW-swimmers have a higher speed in the middle- and long-distance pool swimming events, and to develop a multivariate model that can predict the medalist group in the 10-km competition. Methods: A total of 484 athletes (252-males and 232-females) were included in the analysis. Swimmers were divided into four groups based on their finishing position in the competition. For each swimmer, the absolute best performance (PB) of 200, 400, 800 and 1500-meter in long course, the seasonal best performance (SPB) obtained before the open-water events and critical velocity (CV) were analyzed. Multivariate analysis of variance (MANOVA) was used to detect significant differences between groups and discriminant analysis was used to predict a grouping variable. Results: All the variables analyzed were significantly different between groups (*p* < 0.001). The first discriminant function correctly classified 50% of the overall female and male swimmers. Conclusion: Fastest OW-swimmers have a higher speed in middle- and long-distance pool swimming events. Further studies should include different anthropometric and physiological variables to increase the accuracy of classification.

## 1. Introduction

The international swimming federation (FINA-Fédération Internationale de Natation) defines open-water swimming (OWS) as any competition that takes place in rivers, lakes, oceans, or water channels [1]. Three distances, 5-, 10-, and 25-km are present in World and European championships, while only the 10 km is an Olympic event, and a multi-lap 2500-meter course is usually used [2]. Being a relatively young Olympic discipline, after the admission of OWS in the Olympic Games (Beijing, 2008) numerous swimmers, specialists in middle- and long-distance pool swimming events, started to compete in OWS.

Recent studies have analyzed the pacing adopted during the 10-km race in the main international competitions (European championships, EC; World Championships, WC; Olympic games, OG) [3,4,5,6] showing that it is a common feature for the fastest open-water swimmers (OW-swimmers) to start slowly, adopting a negative pacing strategy (the first half of the race slower than the second) [3,5,6]. Specifically, speed has been shown to significantly increase by ~7%, ~6%, ~4% in the fastest male and female swimmers during the last split of the 5-, 10- and 25- km races, respectively [3]. It has been shown that the most successful 10 km OW-swimmers adopt a conservative starting strategy, which means they swim a large proportion of the race at similar velocities to the rest of their competitors, being in fact positioned in the middle of the main group until the final lap. Thereafter, the ability to change speed at the end of the race and sustain high velocities will differentiate medalists from non-medalists [5].

This strategy highlights the tactical nature of open water races and has inevitably brought to a very high performance density (i.e., difference between first and 10th place or first and last place) compared to disciplines of the same time duration such as a marathon or triathlon [4]. During the Olympic games in Rio 2016, the difference in performance between the 1st and the 10th male and female athlete, was 2.91% and 3.15% for the marathon, 1.92% and 2.28% for triathlon and only 0.07% and 0.81% for the OW-swimmers. In fact, only 5 s divided the first from the 10th athlete [2,7]. Moreover, during the 10-km race at the European Championships in 2016, the female race assigned 2 gold medals, while during the World Championships of Budapest (2017), in the female race four swimmers sprinted in the final 500-meter for the last two medals available, and only one second divided the first three male athletes [2,7].

There is an established subgroup of swimmers that may compete in both open-water and 800- and 1500-meter pool events [8]. Oussama Mellouli, Ferry Weertman, Jordan Wilimovsky, Sharon Van Rouwendaal and Samantha Arevalo are some examples of male and female swimmers that have raced in the pool and OWS events in the same edition of World Championships and Olympic Games. Jordan Wilimovsky and Sharon Van Rouwendaal are able to swim the 800-meter in 7:45.19 (mm:ss.00) and 8:24.12, the 1500-meter in 14:45.03 and 16:03:37, respectively (www.swimrankings.net, accessed on: 24 September 2017). Furthermore, Oussama Mellouli is the first swimmer to win a gold medal in both pool and open-water swimming events in the Olympic Games [9].

Therefore, the first aim of the present work was to determine if the fastest OW-swimmers have a higher speed in middle- and long-distance pool swimming events. The second aim was to develop a multivariate model that can predict the medalist group during the 10-km OWS competition. We hypothesized that the fastest OW-swimmers would have the fastest times in middle- and long-distance pool events. The knowledge of these results could allow the coaches to establish the performance indicators, useful for training plans and competition strategies.

## 2. Materials and Methods

In the analysis, we included the results of the main international 10-km OWS competitions. The official finishing ranking was obtained from the website of Ligue Européenne de Natation (LEN, www.len.eu, accessed on: 24 September 2017), Fédération Internationale de Natation (FINA, www.fina.org, accessed on: 24 September 2017) and International Olympic Committee (IOC, www.olympic.org, accessed on: 24 September 2017). A total of 27 events between 2000 and 2017 were included in the analysis. OG: 2016 in Rio, 2012 in London and 2008 in Beijing; WC: 2017 in Budapest, 2015 in Kazan, 2013 in Barcelona, 2011 in Shanghai, 2010 in Roberval, 2009 in Rome, 2008 in Seville, 2007 in Melbourne, 2006 in Naples, 2005 in Montreal, 2004 in Dubai, 2003 in Barcelona, 2002 in Sharm el-Sheikh, 2001 in Fukuoka and 2000 in Honolulu. EC: 2016 in Hoorn, 2014 in Berlin, 2012 in Piombino, 2011 in Eilat, 2010 in Budapest, 2008 in Dubrovnik, 2006 in Budapest, 2004 in Madrid, 2002 in Berlin.

Only the races where the ranking was available in the official results were included in the analysis. Athletes who did not start or finish (86-males, 70-females) or who were disqualified (16-males, 11-females) were not included in the analysis. A total of 1751 (953-males and 798-females) athletes were identified in the first phase of the analysis.

Swimmers were divided into four groups based on their finishing ranking in the OWS competition. Group 1 (G1; 81-males and 82-females) whose finishing positions were between 1st and 3rd, Group 2 (G2; 189-males and 188-females) whose finishing positions were between 4th and 10th, Group 3 (G3; 468-males and 342-females) whose finishing positions were between 11th and 30th for males and 11th and 24th for females, Group 4 (G4; 215-males and 186-females) whose finishing positions were above the 31st position for males and 25th for females. This subdivision was selected according to a previous study [5].

For each swimmer the absolute best performance (PB) of 200, 400, 800 and 1500-m in long course (50-meter) and the seasonal best performance (SPB) obtained before the open-water events analyzed were retrieved from the official website (www.swimrankings.net, accessed on: 24 September 2017). PB refers to the best result the athlete achieved in his/her career in a single pool swimming race obtained before the open-water event, while SPB refers to the best result obtained in the season of the open-water event analyzed. The final race times of each swimmer were converted in speed (m s^−1^). Critical velocity (CV) of each swimmer was calculated by the slope of the distance-time relationships showed by Wakayoshi et al. [10]. CV has been defined as the highest speed that can be sustained on the basis of the maximal aerobic power [11], and it is considered a valid and practical index of aerobic endurance [12]. The advantage of using the CV is the possibility to collect data during competitions without interfering in training [13]. CV was calculated with possible combinations of two (200-meter and 400-meter; 400-meter and 800-meter; 800-meter and 1500-meter), three (400-meter, 800-meter and 1500-meter) and four (200-meter, 400-meter, 800-meter and 1500-meter) race distances obtained from the SPB and PB of each swimmer, as reported by Toubekis et al. [14] and Zacca et al. [13].

Athletes who did not perform a middle- and long-distance swimming event, or the performance results were not available on the website, were not included in the analysis. A total of 484 athletes (252-males and 232-females) were included in the analysis: G1, 17-males and 23-females; G2, 35-males and 42-females; G3, 143-males and 113-females; G4, 57-males and 54-females. The data collection process is shown by a flow chart in Figure 1. As the data are public and available on the internet, no formal ethics committee approval was necessary.

### Statistical Analysis

Data are presented as mean ± standard deviation (SD). All statistical analysis was performed using the statistical software PASW statistics 25 (SPSS Inc, Chicago, IL, USA). All data were tested for normal distribution using a Kolmogorov-Smirnov test.

Multivariate analysis of variance (MANOVA) was used to detect significant differences between groups. When these were detected a post-hoc Scheffé analysis was performed.

Although different statistical methods applied to sport sciences can be used to determine which predictor variables are related to performance, discriminant analysis has been used in several sport disciplines to determine the different characteristics of athletes in order to better detect “talent” and to finalize training [15,16,17,18,19,20]. Therefore, discriminant analysis was used to determine which variables best distinguished between the different groups. The discriminant analysis derives several equations as a linear combination of independent variables (pool swimming performances), called discriminant functions (DF). The first DF maximizes the differences between the groups in the dependent variable and will be the most powerful to separate the groups while the subsequent DF may not show additional significant differentiation. The magnitudes of the equation coefficients (standardized canonical coefficients) indicate how the discriminating variables affect the score. The higher coefficients indicate which variables have the greatest impact on the discriminant function. The independent variables were entered simultaneously in the analysis. The level of significance was set at *p* ≤ 0.05.

## 3. Results

Table 1 shows differences in the SPB, PB and CV of each group for female swimmers. Table 2 shows the standardized canonical coefficients obtained for each DF provided by the model. The discriminant analysis for female groups showed that DF1 and DF2 were significant (*p* < 0.001 and *p* = 0.026 respectively). However, DF1 showed a higher discriminating power compared to DF2 considering the values of Wilks’ Lambda (lower) and χ^2^ (higher). According to DF1 the most important variables that discriminate the difference between female medalists and the other groups are the 800- and 1500-meter PB (Table 2). The analysis was able to correctly classify only 50.4% of all female swimmers, however, 70% of G1 swimmers were correctly classified.

Table 3 shows differences in the SPB, PB and CV for each group of male swimmers. Table 4 shows the standardized canonical coefficients obtained for each variable in the three DF provided by the model. The discriminant analysis for male groups showed that only DF1 was significant (*p* < 0.001), and the most important variables that discriminate the difference between male medalists and the other groups are the 1500-meter PB and the CV calculated with the SPB of the 400-,800- and 1500-meter (Table 4). The analysis was able to correctly classify only 50% of all male swimmers, however 94.1% of G1 swimmers were correctly classified.

## 4. Discussion

The first aim of the present work was to determine if the fastest open-water swimmers have a higher speed in middle- and long-distance pool swimming events. As hypothesized, the fastest OW-swimmers (G1 and G2) showed significant higher speeds in middle- and long-distance pool events compared to the other groups.

A common pacing strategy observed during OWS competition is to swim behind the leader of the race or the lead pack [3,4,5,6]. This pacing strategy allows to optimize the benefits of drafting and to reduce the energy cost of swimming [5]. Indeed, swimming behind another swimmer at a distance of 0 to 50 cm reduces by 11–38% the metabolic response of the draftee [21]. A sheltered position (in the pack) allows the swimmer to control the race and to save energy for the final end spurt [21]. Previous studies [3,4,5,6] investigating pacing strategy during the 10-km event, showed that the fastest males and females adopt negative pacing, meaning a slow start and a significantly faster last split of the race (2500 m). Successful swimmers have been shown to be prepared not only to maintain but also to increase their swimming pace by at least 1 s per 100 m from the final quarter of the race distance [5].

Although the official split times are every 2.5 km, it is generally accepted (personal communication), that the final end spurt occurs on a much shorter distance. Indeed, the official video of the Olympic race in Rio (2016) [22] showed that the male head group was still together until the last buoy at 350-meter to the arrival and only 5 s divided the first from the 10th athlete [4]. These race tactics are very similar to what previously reported for sprinters in road cycling, where the last 10 min prior to the sprint are the most crucial part of the competition [23]. In this phase of the race, intensity dramatically increases, and cyclists tend to find the best position within the peloton [23]. In fact, similarly to road cycling sprinters, that are required to ride for prolonged periods at moderate intensities for the longest part of the race and have a very demanding final km [23], successful OW-swimmers swim for more than 75% of the race within the pack reducing energy demands, at intensities below average race pace and have a final sprint in the last few hundred meters. It can be assumed that in OWS competitions the aerobic system contribution is prevalent during the majority of the race, however, the large increase in the exercise intensity in the final moments of the race has been related to an anaerobic energy reserve and is considered a typical feature of successful competitors at the end of endurance events who seek for the best possible finishing position [5]. Foster et al. [24] showed that athletes are able to regulate their energetic output over time in a manner designed to optimize performance. Consequently, OW-swimmers, similarly to cyclists, need to find the right moment and the timing to increase the speed at the end of the race. Contrary to other disciplines of similar time duration such as a marathon, where it is difficult to retrieve in the same season times on shorter distance track events, for swimmers it is normal to compete in many pool events throughout the season. The data of the present study showed that both male and female open-water swimmers of G1 and G2 have a higher speed in the middle- and long-distance pool swimming events compared to the other groups. Furthermore, the CV calculated with the PB in 400- and 800-meter was significantly higher in G1 males compared to the other groups (Table 3). We can hypothesize that G1 male swimmers are able to sustain a higher speed (or higher percentage of VO_2max_) for a longer time compared to the swimmers of the other groups.

The second aim was to develop a multivariate model that could predict the medalist group in the 10-km OWS competition. Although discriminant analysis has been used to determine the physiological and anthropometric differences between different types of cyclists, and to develop a multivariate model that can classify and predict the specialty to which emerging cyclists might be best suited or to determine the different physiological and anthropometric characteristics between elite and amateur athletes in cycling [15,19,20], the model of the present study correctly classified only 50% of all subjects. However, the discriminant analysis showed a higher discriminating power to correctly classify the swimmers of G1 (70% and 94.1% for females and males respectively). The standardized canonical coefficients showed that the variable that has the greatest discriminant power between medalists and the other groups in both male and female swimmers is the 1500-meter PB (Table 2 and Table 4). However, for the male groups, the CV calculated from the SPB of 400−800 1500 has a great discriminant power while this was not the case for the females. We can speculate that for male swimmers it is more important to be faster in all three pool distances in the same season of the open-water race because more athletes arrive simultaneously close to the finish line of the open water race.

The data of the present study showed that the fastest OW-swimmers have a higher speed in middle- and long-distance pool swimming events confirming the hypothesis that OW-swimmers need to be tactically prepared, with a good aerobic fitness but also with a good anaerobic reserve for a final end spurt. Moreover, the higher speed of the fastest swimmers most probably indicates a higher propelling efficiency compared to the other groups. A swimmer with a good propelling efficiency reduces energy cost and will be able to swim faster than swimmers with poor propelling efficiency and a large metabolic power [25]. The seasonal results in middle- and long-distance pool swimming events may provide useful information regarding the fitness of the athletes during the season.

A limitation of this model was the absence of anthropometric and physiological characteristics in the analysis. It is known that maximal oxygen uptake, lactate threshold and efficiency have an overall impact on endurance performance. The integration of these variables in a successive model may increase the accuracy of classification and predict the medalist group in the 10-km OWS competition.

## 5. Conclusions

In conclusion, the main finding of the present study indicated that both male and female fastest OW-swimmers have a higher speed in middle- and long-distance pool swimming events.

Some of the variables that showed no significant differences between the four groups in the MANOVA had high discriminating power in this model as already shown for example by Peinado et al. [15] in cyclists. The variables that have the greatest discriminant power between the medalists and the other groups for female swimmers are the 800- and 1500-meter PB, whereas for male swimmers are the 1500-meter PB and the CV calculated with the SPB of the 400-,800- and 1500-meter.

The second finding was that the speed in the 1500-meter may be a good predictor to determine the results in both male and female 10-km OWS competitions and the seasonal results in middle- and long-distance pool swimming events may provide useful information regarding the fitness of the athletes during the season, providing useful information for the coaches regarding the orientation of training and the selection of talented OW-swimmers.

Finally, integrating different anthropometric and physiological variables should increase the accuracy of classification to validate and improve the model.

## Figures and Tables

**Figure 1 jfmk-04-00015-f001:**
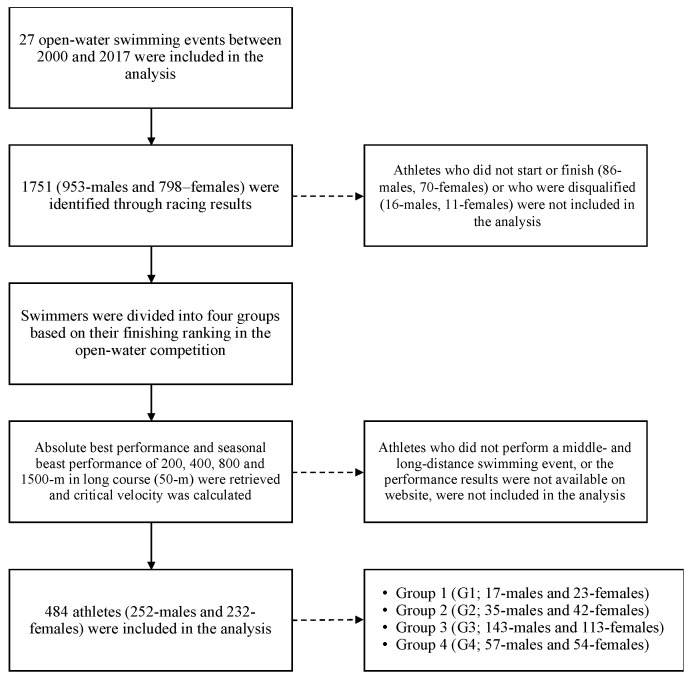
Flow chart of the study. Solid and dotted arrows represent the main and secondary phases of the data collection process respectively.

**Table 1 jfmk-04-00015-t001:** The mean speed (m s^−1^) of PB, SPB and CV of female swimmers. Differences with G3 and G4 are shown as *. Differences with G4 are shown as ^†^.

	G1	G2	G3	G4	F	*p*
PB-200	1.60 ± 0.06	1.62 ± 0.04 *	1.59 ± 0.04	1.58 ± 0.04	F(3,232) = 7.40	<0.001
PB-400	1.58 ± 0.04 *	1.57 ± 0.04 *	1.55 ± 0.03	1.54 ± 0.04	F(3,232) = 12.50	<0.001
PB_800	1.55 ± 0.03 *	1.55 ± 0.03 *	1.52 ± 0.04	1.50 ± 0.04	F(3,232) = 20.22	<0.001
PB-1500	1.53 ± 0.02 *	1.52 ± 0.03 *	1.49 ± 0.04	1.47 ± 0.05	F(3,232) = 21.23	<0.001
SPB-200	1.58 ± 0.06	1.58 ± 0.04 ^†^	1.56 ± 0.04	1.55 ± 0.04	F(3,232) = 5.48	0.001
SPB-400	1.56 ± 0.04 *	1.55 ± 0.03 *	1.53 ± 0.03	1.52 ± 0.04	F(3,232) = 12.73	<0.001
SPB-800	1.54 ± 0.03 *	1.52 ± 0.03 *	1.50 ± 0.04	1.48 ± 0.04	F(3,232) = 17.49	<0.001
SPB-1500	1.51 ± 0.02 *	1.50 ± 0.03 *	1.47 ± 0.04	1.45 ± 0.05	F(3,232) = 19.85	<0.001
CV-PB-200-400	1.55 ± 0.03 *	1.53 ± 0.03 *	1.50 ± 0.04	1.49 ± 0.04	F(3,232) = 13.87	<0.001
CV-PB-400-800	1.53 ± 0.03 *	1.52 ± 0.03 *	1.49 ± 0.05	1.46 ± 0.05	F(3,232) = 19.78	<0.001
CV-PB-800-1500	1.50 ± 0.02 *	1.48 ± 0.03 *	1.46 ± 0.04	1.44 ± 0.06	F(3,232) = 16.54	<0.001
CV-PB-400-800-1500	1.51 ± 0.02 *	1.49 ± 0.03 *	1.47 ± 0.04	1.45 ± 0.05	F(3,232) = 20.01	<0.001
CV-PB-All	1.51 ± 0.02 *	1.50 ± 0.03 *	1.47 ± 0.04	1.45 ± 0.05	F(3,232) = 19.41	<0.001
CV-SPB-200-400	1.54 ± 0.05 *	1.51 ± 0.03 *	1.49 ± 0.05	1.48 ± 0.05	F(3,232) = 11.43	<0.001
CV-SPB-400-800	1.52 ± 0.03 *	1.49 ± 0.03 *	1.47 ± 0.04	1.45 ± 0.05	F(3,232) = 17.00	<0.001
CV-SPB-800-1500	1.48 ± 0.03 *	1.47 ± 0.03 *	1.44 ± 0.05	1.42 ± 0.06	F(3,232) = 15.49	<0.001
CV-SPB-400-800-1500	1.50 ± 0.02 *	1.48 ± 0.03 *	1.45 ± 0.04	1.43 ± 0.05	F(3,232) = 18.65	<0.001
CV-SPB-All	150 ± 0.02 *	1.48 ± 0.03 *	1.45 ± 0.04	1.44 ± 0.05	F(3,232) = 19.63	<0.001

ALL, four (200-meter, 400-meter, 800-meter and 1500-meter); PB, personal best performance; SPB, seasonal best performance.

**Table 2 jfmk-04-00015-t002:** Standardized canonical coefficients for each variable in the three discriminant functions (DF) of female swimmers.

	DF 1	DF 2	DF 3
PB-200	−3.06	1.39	−1.40
PB-400	2.37	−2.14	3.78
PB_800	3.17	−0.02	−1.61
PB-1500	4.74	2.88	−4.23
SPB-200	0.36	0.77	0.42
SPB-400	−0.19	−1.67	−1.41
SPB-800	0.36	−1.36	−0.40
SPB-1500	−3.13	3.53	5.28
CV-PB-200-400	−1.99	0.55	−1.30
CV-PB-400-800	−2.03	−0.19	3.89
CV-PB-800-1500	−0.52	−2.09	6.13
CV-PB-400-800-1500	1.50	−1.84	−3.98
CV-PB-All	−4.86	2.39	−0.43
CV-SPB-200-400	0.45	0.64	0.24
CV-SPB-400-800	0.73	−0.24	0.75
CV-SPB-800-1500	1.09	1.10	0.21
CV-SPB-400-800-1500	1.04	−3.92	−1.16
CV-SPB-All	0.29	0.60	−4.13

DF1: Wilks’ Lambda = 0.52 (χ^2^ = 143.72; *p* < 0.001); DF2: Wilks’ Lambda = 0.79 (χ^2^ = 51.79; *p* = 0.026); DF3: Wilks’ Lambda = 0.95 (χ^2^ = 11.20; *p* = 0.797).

**Table 3 jfmk-04-00015-t003:** The mean speed (m s^−1^) of PB, SPB and CV of male swimmers. Differences with G3 and G4 are shown as *. Differences with G4 are shown as ^†^. Differences with the others ^a^.

	G1	G2	G3	G4	F	P
PB-200	1.80 ± 0.06 *	1.76 ± 0.05 *	1.74 ± 0.06	1.72 ± 0.05	F(3,252) = 13.28	<0.001
PB-400	1.74 ± 0.04 *	1.71 ± 0.04 *	1.67 ± 0.05	1.65 ± 0.05	F(3,252) = 17.67	<0.001
PB_800	1.70 ± 0.03 ^a^	1.65 ± 0.03	1.62 ± 0.05	1.60 ± 0.05	F(3,252) = 21.21	<0.001
PB-1500	1.66 ± 0.03 *	1.63 ± 0.03 *	1.60 ± 0.05	1.58 ± 0.05	F(3,252) = 20.85	<0.001
SPB-200	1.78 ± 0.04 *	1.74 ± 0.05 ^†^	1.71 ± 0.06 ^†^	1.68 ± 0.06	F(3,252) = 14.53	<0.001
SPB-400	1.71 ± 0.04 *	1.69 ± 0.04 *	1.65 ± 0.06	1.63 ± 0.05	F(3,252) = 17.48	<0.001
SPB-800	1.67 ± 0.04 ^a^	1.63 ± 0.04 *	1.59 ± 0.05	1.58 ± 0.05	F(3,252) = 23.24	<0.001
SPB-1500	1.65 ± 0.03 *	1.62 ± 0.03 *	1.57 ± 0.05	1.56 ± 0.05	F(3,252) = 22.10	<0.001
CV-PB-200-400	1.68 ± 0.04 *	1.65 ± 0.04 *	1.62 ± 0.06	1.60 ± 0.06	F(3,252) = 13.81	<0.001
CV-PB-400-800	1.66 ± 0.03 ^a^	1.60 ± 0.03 *	1.57 ± 0.06	1.56 ± 0.05	F(3,252) = 18.90	<0.001
CV-PB-800-1500	1.62 ± 0.03 *	1.61 ± 0.03 *	1.57 ± 0.05	1.55 ± 0.05	F(3,252) = 17.11	<0.001
CV-PB-400-800-1500	1.64 ± 0.03 *	1.61 ± 0.03 *	1.57 ± 0.05	1.56 ± 0.05	F(3,252) = 19.48	<0.001
CV-PB-All	1.64 ± 0.03 *	1.61 ± 0.03 *	1.58 ± 0.05	1.56 ± 0.05	F(3,252) = 19.39	<0.001
CV-SPB-200-400	1.65 ± 0.06 *	1.65 ± 0.05 *	1.59 ± 0.07	1.58 ± 0.06	F(3,252) = 12.20	<0.001
CV-SPB-400-800	1.64 ± 0.04 *	1.58 ± 0.05 *	1.55 ± 0.06	1.53 ± 0.06	F(3,252) = 19.11	<0.001
CV-SPB-800-1500	1.62 ± 0.04 *	1.60 ± 0.04 *	1.55 ± 0.06	1.54 ± 0.06	F(3,252) = 15.42	<0.001
CV-SPB-400-800-1500	1.63 ± 0.04 *	1.59 ± 0.04 *	1.55 ± 0.05	1.53 ± 0.06	F(3,252) = 20.66	<0.001
CV-SPB-All	1.63 ± 0.03 *	1.60 ± 0.03 *	1.55 ± 0.05	1.54 ± 0.06	F(3,252) = 21.35	<0.001

ALL, four (200-meter, 400-meter, 800-meter and 1500-meter); PB, personal best performance; SPB, seasonal best performance.

**Table 4 jfmk-04-00015-t004:** Standardized canonical coefficients for each variable in the three discriminant functions (DF) of male swimmers.

	DF 1	DF 2	DF 3
PB-200	−1.73	1.31	0.13
PB-400	1.89	−3.92	−1.56
PB_800	1.30	1.71	2.71
PB-1500	2.32	−0.88	1.47
SPB-200	0.98	−0.26	2.80
SPB-400	−0.98	2.13	−2.27
SPB-800	0.13	−3.15	−4.83
SPB-1500	−0.97	2.52	0.14
CV-PB-200-400	−1.43	1.98	0.49
CV-PB-400-800	0.17	−0.49	0.67
CV-PB-800-1500	0.96	2.72	3.12
CV-PB-400-800-1500	−0.36	−0.02	−3.51
CV-PB-All	−3.28	−1.89	−2.65
CV-SPB-200-400	0.79	−0.21	2.81
CV-SPB-400-800	−0.35	3.56	4.46
CV-SPB-800-1500	−1.39	3.96	2.37
CV-SPB-400-800-1500	2.70	−7.01	−0.93
CV-SPB-All	0.02	−0.52	−3.37

DF1: Wilks’ Lambda = 0.638 (χ^2^ = 107.76; *p* < 0.001); DF2: Wilks’ Lambda = 0.85 (χ^2^ = 37.85; *p* = 0.298); DF3: Wilks’ Lambda = 0.95 (χ^2^ = 13.26; *p* = 0.653).

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
