# Peer review of "Do the Fastest Open-Water Swimmers have A Higher Speed in Middle- and Long-Distance Pool Swimming Events?"

_jfmk, 2019, doi:10.3390/jfmk4010015_

Reviewer 1 Report

Thank you for your interesting paper.

My key comment is that, although you have (correctly, in my view) done separate analyses for males and females you have done no discussion, at all, as to why the results for the different genders may differ. The justification for the paper, as written, also seems to rely too much on anecdotal information (e.g. as regards performance densities) as opposed to data. Moreover, although  it is very interesting that pacing strategies in different disciplines with the same overall duration differ (triathlon, cycling, swimming) you do not go into the wider implications of this, and your results, in the discussion. I think that if you did so you might increase the interest of your paper, to a larger audience. Moreover, it could be sensible to include some justification as to the extent to which the methods and statistics that you used were fit for purpose. 

The manuscript would also benefit from further careful proof reading for missing words (e.g. "as" in line 33, "the" in lines 103 and 109), inconsistent or incorrect spelling (e.g. you write "analyzed" in the abstract and "analysed" in line 36 of the text, and "beast" instead of "best" in line 92), formatting (e.g. you put an extra space on lines 39, 42, 45, and 97; you have a one sentence paragraph in lines 54-56; you put a comma rather than a full stop at the end of line 79) and grammar (e.g. the sentences from lines 44-46, 138-139, and 172-173 are ungrammatical). I also think that your abstract could be better phrased- it does not make your work "shine" as much as it could.

Congratulations, however, on your command of English. I do realise that English is not your first language and am impressed at how well you write in it. The language related comments that I made can be easily addressed.

Specific comments are given below:

L42-44 Consider describing the results of this study in more detail- given that it is the justification for this paper. 

L45-47 Consider making it more clear as to when you are referring to male, and when to female, athletes. The performance density for them in certainly different in triathlon.

L65-69 Please avoid one sentence paragraphs.

L91- best ever (over how long?- as it has implications for the relevance) and best for which season? This is unclear. 

L99-101 Consider explaining this further. 

Good that you included Fig 1 to clarify the data selection process- not that easy to follow how many people were in what from just the text.

L172-175 This is anecdotal. Is this statement based on only one race?

L184-187 So? Please avoid these leading one sentence paragraphs. They break up the text and make the reader more likely to lose the thread of what you are trying to get across.

Fig 2 to me, in laymans language, is as clear as mud. Nor am I sure how much of it would be decipherable at page size. You mention DF1 but I am unsure as to which you mean - is this the first of the CV calculations and the 800-1500m one of that? Statistics is not my strong point but even bearing that in mind I do not think you have made your methodology sufficiently clear.

Warm regards

Author Response

Author’s response to the reviewers’ comments

 We are grateful to the Editor for giving us the possibility to revise the manuscript and making it suitable to be considered for publication. The reviewers’ comments proved exceptionally helpful in the revision process. The manuscript was revised in structure and content according to their suggestions. Each comment was carefully addressed with a specific answer (R) and the respective changes were inserted in the manuscript.

Reviewer 1
My key comment is that, although you have (correctly, in my view) done separate analyses for males and females you have done no discussion, at all, as to why the results for the different genders may differ. The justification for the paper, as written, also seems to rely too much on anecdotal information (e.g. as regards performance densities) as opposed to data. Moreover, although  it is very interesting that pacing strategies in different disciplines with the same overall duration differ (triathlon, cycling, swimming) you do not go into the wider implications of this, and your results, in the discussion. I think that if you did so you might increase the interest of your paper, to a larger audience. Moreover, it could be sensible to include some justification as to the extent to which the methods and statistics that you used were fit for purpose.

R- We thank the reviewer for the positive feedback and responded point to point to his/her useful suggestions.  We added in the discussion possible explanations for the differences observed between males and females.

The performance density is not an anecdotal information but based on specific research (Zingg et al 2014, Rüst et al, 2014, Baldassarre et al 2017) and we do believe that part of these differences are due to differences in performance densities between male and female races. Moreover, we reported and described several race situations to have a match on what happened on field and the paper results. These data are useful and add value to the results we found. Different performance density indicates different ways of racing for males and females and for different disciplines. We did not go too much in detail on the differences with the other disciplines (i.e. triathlon and marathon) because it is not possible to perform the same analysis as explained in the text and because it was not the purpose of the paper. More information on this topic are reported in the papers of Baldassarre et al., 2018.

We have included some justification and references on the methods and statistics used.

-          Zingg, M.A.; Rüst, C.A.; Rosemann, T.; Lepers, R.; Knechtle, B. Analysis of sex differences in open-water ultra-distance swimming performances in the FINA World Cup races in 5 km, 10 km and 25 km from 2000 to 2012. BMC Sport. Sci. Med. Rehabil. 2014, 6, 7.

-          Rüst, C.A.; Lepers, R.; Rosemann, T.; Knechtle, B. Will women soon outperform men in open-water ultra-distance swimming in the “Maratona del Golfo Capri-Napoli”? Springerplus 2014, 3, 86.

-          Baldassarre, R.; Bonifazi, M.; Zamparo, P.; Piacentini, M.F. Characteristics and Challenges of Open-Water Swimming Performance: A Review. Int. J. Sports Physiol. Perform. 2017, 12, 1275–1284. 1.

-          Baldassarre, R.; Bonifazi, M.; Piacentini, M.F. Pacing profile in the main international open-water swimming competitions. Eur. J. Sport Sci. 2018, (In press), 1–10.

The manuscript would also benefit from further careful proof reading for missing words (e.g. "as" in line 33, "the" in lines 103 and 109), inconsistent or incorrect spelling (e.g. you write "analyzed" in the abstract and "analysed" in line 36 of the text, and "beast" instead of "best" in line 92), formatting (e.g. you put an extra space on lines 39, 42, 45, and 97; you have a one sentence paragraph in lines 54-56; you put a comma rather than a full stop at the end of line 79) and grammar (e.g. the sentences from lines 44-46, 138-139, and 172-173 are ungrammatical). I also think that your abstract could be better phrased- it does not make your work "shine" as much as it could. Congratulations, however, on your command of English. I do realise that English is not your first language and am impressed at how well you write in it. The language related comments that I made can be easily addressed.

R- These corrections were done according to the suggestions, and the abstract has been modified. Specific attention was paid to the grammar and language mistakes

Specific comments are given below:

L42-44 Consider describing the results of this study in more detail- given that it is the justification for this paper.

R- The statistical method utilized was described better. The results and tables now are clearer for the reader. We added some more details in the results section.

L45-47 Consider making it more clear as to when you are referring to male, and when to female, athletes. The performance density for them in certainly different in triathlon.

R- We have included the specific performance density in male and female athletes in marathon, triathlon and open-water swimming.

L65-69 Please avoid one sentence paragraphs.

R- The sentence has been united in the previous paragraph.

L91- best ever (over how long?- as it has implications for the relevance) and best for which season? This is unclear.

R- The best performance is referring to the best performance ever before the open-water race. An example, if an athlete performed the best performance in 200-m in 2009 and the same athlete performed the open-water race in Rio 2016, we considered in the analysis the time obtained in 2009 as the absolute best performance in the 200-m.

While the seasonal best performance is referring to the single season when open-water race was performed. An example, if an athlete performed the open-water race in London 2012, we considered in the analysis the times obtained in pool races prior to the Olympic race.

L99-101 Consider explaining this further. 

R- Critical velocity was calculated from possible combinations of two, three and four timed distanced as reported by Wakayoshi et al 1996, Toubekis et al 2006 and Zacca et al 2010.

-          Wakayoshi, K.; Yoshida, T.; Kasai, T.; Moritani, T.; Mutoh, Y.; Miyashita, M. Validity of critical velocity as swimming fatigue threshold in the competitive swimmer. Ann. Physiol. Anthropol. 1992, 11, 301–307.

-          Toubekis, A.G.; Tsami, A.P.; Tokmakidis, S.P. Critical Velocity and Lactate Threshold in Young Swimmers. Int. J. Sports Med. 2006, 27, 117–123.

-          Zacca, R.; Wenzel, B.M.; Piccin, J.S.; Marcilio, N.R.; Lopes, A.L.; De Souza Castro, F.A. Critical velocity, anaerobic distance capacity, maximal instantaneous velocity and aerobic inertia in sprint and endurance young swimmers. Eur. J. Appl. Physiol. 2010, 110, 121–131.

Good that you included Fig 1 to clarify the data selection process- not that easy to follow how many people were in what from just the text.

R- We have reported in the text the number of athletes selected in each phases of the study and in the last part of the "Materials and Methods" section we have reported the final number of athletes involved in the analysis. Figure 1 reports step by step the data collection process and it helps the reader to understand the methods used.

L172-175 This is anecdotal. Is this statement based on only one race?

R- It is generally accepted by experience and video footage that in the last meters of the race there is an important increase of speed. We reported a single real race data to have a match on what happened on field. However the sentence has been modified.

L184-187 So? Please avoid these leading one sentence paragraphs. They break up the text and make the reader more likely to lose the thread of what you are trying to get across.

R- The sentence has been modified.

Fig 2 to me, in laymans language, is as clear as mud. Nor am I sure how much of it would be decipherable at page size. You mention DF1 but I am unsure as to which you mean - is this the first of the CV calculations and the 800-1500m one of that? Statistics is not my strong point but even bearing that in mind I do not think you have made your methodology sufficiently clear.

R- Thank you for your useful comment. Figure 2 and 3 have been removed. A focus on statistical methods was added in the “Materials and Methods” paragraph with several references. The discriminant analysis derives several equations as linear combination of independent variables (pool swimming performances), called discriminant functions (DF). The first DF maximizes the differences between the groups in the dependent variable and will be the most powerful to separate the groups while the subsequent DF may not show additional significant differentiation. The subsequent DFs are uncorrelated with the first function and maximize the differences between the groups. The magnitudes of the equation coefficients (standardized canonical coefficients) indicate how the discriminating variables affect the score. The higher coefficients indicate which variables have a greatest impact on the discriminant function. DF is referring to first discriminant functions that allows to determine which variable(s) are the best predictors used in the model.

Reviewer 2 Report

The present study seeks to determine if the fastest open water swimmers have a higher speed in the middle distance pool swimming events. This is an interesting question for the open water community and for the elite level coaches and swimmers. 

The manuscript is adequately structured, the scientific language is of a good standard and it presents results in a proper form. However, the present reviewer feels some questions could be more precisely explained in order to maximize the potential of the present research:

First of all, authors should modify the phrasing of the first aim at the end of introduction as they did not measure the "unique ability to accelerate at the end of a 2h race". Therefore, it is strongly recommended to present the first aim as in the abstract section. 

Also, in the introduction section, it is recommended to provide some specific information about the swimming paces and the changes in velocity that explain the key points brought by authors (such as the velocity spurts at the end of 10 m races, the personal best times of some of the elite swimmers they mention, specific data about negative race strategy observed during major competitions, …etc.).

The writing of the results section could be improved in order to highlight the key aspects of the tables/figures presented. Some important data like the level of the pool swimming events could be expressed in seconds to ease the understanding of readers. Also, in the methods section, the paragraph structure (ten paragraphs in the actual form) could be improved. 

Finally, both in the discussion and conclusion sections the authors state that "the fastest OW
swimmers showed significant higher speeds in middle- and long-distance pool events compared to the other groups". However, results do not show clear differences in middle- and long-distance pool PB's between G1 and G2. Therefore, from the point of view of the present reviewer, authors should highlight that the 800 and 1500m PB's seem to be necessary conditions to opt to the G1 and G2 performance groups. However, to obtain fast personal records in these events does not seem to be enough to achieve success in the open water events. This seems an important point that should be expressed accordingly.

Some other minor corrections should be:

- Authors should delete extra-spaces in lines 39, 42, 45 and 97

- Paragraphs in lines 54 and 57 should be merged. 

- Lines 102-103 should be acknowledge as an important limitation of the study. It would be interesting to know how many swimmers from G1 and G2 do not present middle- and long-distance pool results. 

- Paragraph structure in the first half of the discussion section should be improved.

Author Response

Authors' Responses to Reviewer 2's Comments

We are grateful to the Editor for giving us the possibility to revise the manuscript and making it suitable to be considered for publication. The reviewers’ comments proved exceptionally helpful in the revision process. The manuscript was revised in structure and content according to their suggestions. Each comment was carefully addressed with a specific answer (R) and the respective changes were inserted in the manuscript.

Reviewer 2

The present study seeks to determine if the fastest open water swimmers have a higher speed in the middle distance pool swimming events. This is an interesting question for the open water community and for the elite level coaches and swimmers.
The manuscript is adequately structured, the scientific language is of a good standard and it presents results in a proper form. However, the present reviewer feels some questions could be more precisely explained in order to maximize the potential of the present research:

First of all, authors should modify the phrasing of the first aim at the end of introduction as they did not measure the "unique ability to accelerate at the end of a 2h race". Therefore, it is strongly recommended to present the first aim as in the abstract section.

R- We thank the reviewer for the interest in our work responded point to point to his/her useful suggestions. The first aim was modified according the suggestions.

Also, in the introduction section, it is recommended to provide some specific information about the swimming paces and the changes in velocity that explain the key points brought by authors (such as the velocity spurts at the end of 10 m races, the personal best times of some of the elite swimmers they mention, specific data about negative race strategy observed during major competitions, …etc.).

R-We included in the introduction some information about the pacing during open-water races and specific data about negative race strategy. The personal best times of some of the elite swimmers was included.

The writing of the results section could be improved in order to highlight the key aspects of the tables/figures presented. Some important data like the level of the pool swimming events could be expressed in seconds to ease the understanding of readers. Also, in the methods section, the paragraph structure (ten paragraphs in the actual form) could be improved.

R- The results section presents the data without bias or interpretation. The data are shown in a logical sequence without duplicate the results between the text and the figures or table. We have decided to use the speed to allow a direct comparison with previous studies and because in science the standard unit of measure for speed is generally meters per second. The method section was revised.

Finally, both in the discussion and conclusion sections the authors state that "the fastest OW swimmers showed significant higher speeds in middle- and long-distance pool events compared to the other groups". However, results do not show clear differences in middle- and long-distance pool PB's between G1 and G2. Therefore, from the point of view of the present reviewer, authors should highlight that the 800 and 1500m PB's seem to be necessary conditions to opt to the G1 and G2 performance groups. However, to obtain fast personal records in these events does not seem to be enough to achieve success in the open water events. This seems an important point that should be expressed accordingly.

R-According MANOVA there is not a clear difference between G1 and G2. However, MANOVA was used to detect the single significant differences between groups. Some of the variables that showed no significant differences between the four groups in the MANOVA had high discriminating power in this model as already shown for example by Peinado et al. (2011). Whereas, the discriminant analysis creates a model to discriminate the groups considering the variables together. The variables that have a greatest discriminant power between the medalists and the other groups for female swimmers are the 800- and 1500-m PB, whereas for male swimmers are the 1500-m PB and the CV calculated with the SPB of the 400-, 800- and 1500-m.

Some other minor corrections should be:

- Authors should delete extra-spaces in lines 39, 42, 45 and 97

R- These corrections were done according to the suggestions.

- Paragraphs in lines 54 and 57 should be merged.

R- The paragraphs were merged.

- Lines 102-103 should be acknowledge as an important limitation of the study. It would be interesting to know how many swimmers from G1 and G2 do not present middle- and long-distance pool results.

R- We reported in the text and in Figure 1 the number of athletes selected in each phases of the study therefore the final number of athletes involved in the analysis. The number of swimmers that do not present all middle- and long-distance pool results is 1267 (701-males and 566-females): G1, 64-males and 59-females; G2, 154-males and 146-females; G3, 325-males and 229-females; G4, 158-males and 132-females. However we did not include this in the text

- Paragraph structure in the first half of the discussion section should be improved.

R- The discussion section was modified according the suggestions of both reviewers.

Round  2

Reviewer 2 Report

Authors have adequately responded to the questions raised by the present reviewer.